# Evaluation of the articular cartilage in the knees of rats with induced arthritis treated with curcumin

Tiago Nicoliche[1], Diogo Correa Maldonado[1], Jean Faber[2], Marcelo Cavenaghi Pereira da Silva[1]*

1 Department of Morphology and Genetics, Universidade Federal de São Paulo, São Paulo, São Paulo, Brazil, 2 Department of Neurology and Neurosurgery, Universidade Federal de São Paulo, São Paulo, São Paulo, Brazil

* marcaven@gmail.com

**Data Availability Statement:** All relevant data are within the paper and its Supporting Information files.

**Funding:** TN was financed by Coordenação de Aperfeiçoamento de Pessoal de Nível Superior -

## Abstract

This study was designed to evaluate the anti-inflammatory effects of a curcumin treatment on the knee of rats with induced osteoarthritis. Fifteen adult rats were used and divided in three groups: the osteoarthritis group (OAG), control group (CG–without induction of osteoarthritis), and curcumin-treated osteoarthritis group (COAG). Osteoarthritis was induced in the right knee of rats in the OAG and COAG by administering an intra-articular injection of 1 mg of zymosan. Fourteen days after induction, 50 mg/kg curcumin was administered by gavage daily for 60 days to the COAG. After the treatment period, rats from all groups were euthanized. Medial femoral condyles were collected for light microscopy and immunohistochemical staining. The expression of SOX-5, IHH, MMP-8, MMP-13, and collagen 2 (Col2) was analyzed. The COAG exhibited an increase in the number of chondrocytes in the surface and middle layers compared with that of the OAG and CG, respectively. The COAG also showed a decrease in the thicknesses of the middle and deep layers compared with those of the OAG, and an increase in Col2 expression was observed in all articular layers (surface, middle, and deep) in the COAG compared with that in the OAG. SOX-5 expression was increased in the surface and deep layers of the COAG compared with those in the OAG and CG. Based on the results of this study, the curcumin treatment appeared to exert a protective effect on cartilage, as it did not result in an increase in cartilage thickness or in MMP-8 and MMP-13 expression but led to increased IHH, Col2, and SOX-5 expression and the number of chondrocytes.

## Introduction

Osteoarthritis (OA) is one of the most burdensome diseases worldwide, and it is caused by a metabolic imbalance due to extrinsic factors, such as trauma, or intrinsic factors, such as various types of molecular dysfunction. The imbalance induces the production of a cascade of cytokines and inflammatory mediators that results in the release of nitric oxide (NO), leading to chondrocyte apoptosis and extracellular matrix (ECM) degeneration [1].

Brasil (CAPES) - Finance Code 001 and MCPS by grant #2018/11235-2, São Paulo Research Foundation (FAPESP). The funders had no role in study design, data collection and analysis, decision to publish, or preparation of the manuscript.

**Competing interests:** The authors have declared that no competing interests exist.

During the articular inflammatory process, macrophages release some cytokines, such as interleukin (IL)-1β, IL-6, IL-8, and tumor necrosis factor (TNF)-α, as well as many growth factors. These cytokines induce the expression of matrix metalloproteinases (MMPs) [2] that digest both macromolecules of the ECM and molecules that are not located in the ECM, such as receptors, growth factors, cytokines, and chemokines [3].

The collagen degeneration in OA cartilage gradually spreads around the chondrocytes according to the cartilage layers, beginning in the surface layer and extending to the deep layer [4]. The MMPs functioning as collagenases are involved in the first step of collagen cleavage. MMPs 1, 8, and mainly MMP-13 are the MMPs responsible for the Col2 cleavage next to the extreme C-terminus [3,5].

Some transcription factors, such as the SOX trio (SOX-5, SOX-6, and SOX-9), are crucial for the development of primordial cartilage, formation of the growth plate and maintenance of the chondrogenesis phenotypes [6]. The SOX trio is coexpressed in all primordial cartilage, and specifically, SOX-9 directly induces the expression of the Col2 gene [7].

In addition, during limb development, Indian hedgehog (IHH) is initially detected in the precartilaginous condensation mesenchyme when Sonic hedgehog (Shh) signaling becomes weaker [8]. Then, IHH is broadly expressed throughout the cartilage and subsequently becomes limited to prehypertrophic chondrocytes [9]. IHH stimulates the formation of the intramembranous bone in the periosteum next to the diaphysis and in endochondral ossification; it regulates chondrocyte hypertrophy [10] and stimulates the replacement of the calcified cartilage matrix by bone in the metaphysis [11].

Considering the human diet and available treatments for inflammatory diseases, turmeric is used in Ayurveda and Chinese medicine due to its antioxidant, antimicrobial and anti-inflammatory properties [12–14]. Curcumin, which is the active principle component of turmeric, alters the inflammatory response by decreasing the levels of cyclo-oxygenase-2 (COX-2), lipo-oxygenase, and induced nitric oxide synthase (iNOS). In addition, curcumin inhibits the production of inflammatory cytokines, such as, TNF-α, IL-1, -2, -6, -8, and -12; monocyte chemoattractant protein (MCP); and migration inhibitory protein, and decreases the regulation of mitogenic activation and Janus kinase (JAK) [12] (Fig 1).

Based on this information, the purpose of the present study is to confirm the efficacy of a curcumin treatment in an animal model of induced osteoarthritis using a histomorphometric method and immunohistochemical staining for proteins related to the articular structure and osteoarthritis development.

## Materials and methods

### Experimental design

The animal procedures were approved by Institutional Care and Use Committee (IACUC) of Federal University of São Paulo (Animal Care Ethical Committee approval number 1918020617).

Fifteen male Wistar rats (*Rattus norvegicus*) that were bred and housed until they were aged 12 weeks in CEDEME (Central de Desenvolvimento de Modelos Experimentais para Biologia e Medicina) of Federal University of São Paulo on a natural dark and light (12:12 h) cycle and water and standard food available *ad libitum* were used in the present study. The animals were randomly distributed into three groups (n = 5 animals each): the control group (CG): without induction of OA and treated with a saline solution; the osteoarthritis group (OAG): without treatment; and the curcumin-treated osteoarthritis group (COAG): animals with osteoarthritis that were treated with curcumin.

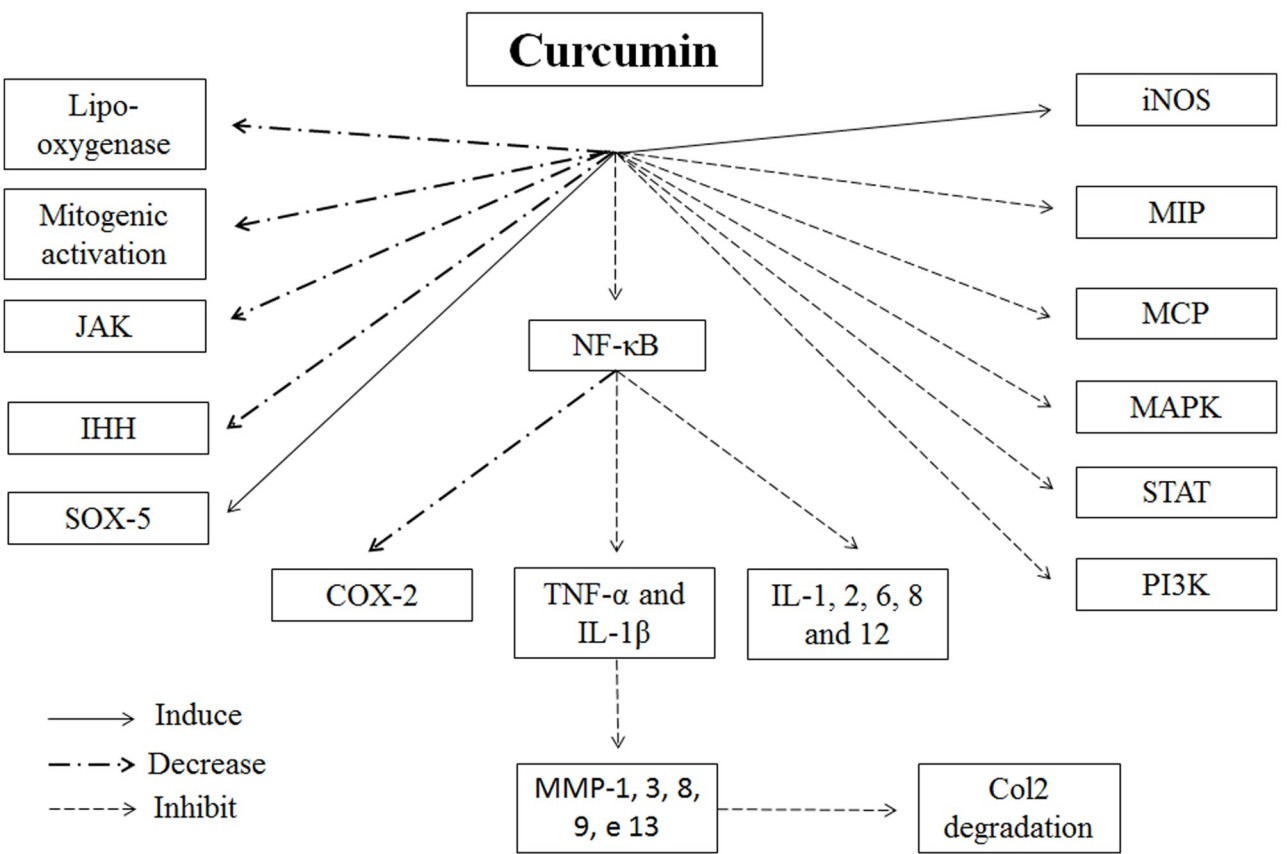

**Fig 1. Diagrammatic representation of the curcumin role on inflammation and Col2 degradation.** The curcumin decreases lipo-oxygenase, mitogenic activation, JAK (Janus Kinase), IHH (Indian hedghog) and also COX-2 (cyclo-oxygenase-2) via NF-B inhibition. The curcumin induces SOX-5 and iNOS (induced nitric oxide synthase). The curcumin inhibit MAPK (mitogen-activated protein kinase), MCP (monocyte chemoattractant protein), STAT (signal transduction and activation transcription), PI3K (phosphoinositide 3-kinase), MIP (Migration inhibitory protein). The expression of IL-1, 2, 6, 8 and 12 (interleukin), TNF-α (Tumour Necrosis Factor alpha) and IL-1β are inhibited via NF-B (nuclear factor kappa B) inhibition. The MMP-1, 3, 8, 9 and 13 (matrix metalloproteinases) are mainly inhibited via TNF-α and IL-1β inhibition which leads to an inhibition in Col2 degradation.

## Osteoarthritis induction and treatment

Under anesthesia (ketamine—Dopalen, Sespo indústria e comércio, São Paulo, Brazil: 100 mg/kg and xylazine—Calmium, União química farmaceutica, São Paulo, Brazil: 10 mg/kg), osteo-arthritis was induced in the right knees of rats in the OAG and COAG by administering an intra-articular injection of a single dose of 1 mg of zymosan (Sigma-Aldrich CAS number 58856-93-2) dissolved in 50 μL of sterile saline [15]. After trichotomy and antiseptic treatment with povidone, a needle was carefully inserted in the right knee just below the femoro-patellar ligament using a medial approach. Thereafter, the animals rested for 14 days to allow inflam-mation and chondral degeneration to occur. The same induction procedures were performed in the CG by applying sterile saline. The animals were daily monitored for abscesses and amount of food and water consumption.

Fourteen days after induction, 50 mg/kg curcumin (BIOVEA® CURCUMIN bcm-95, Scottsdale, Arizon, USA) were administered by gavage to the COAG daily for 60 days. The CG and OAG were administered sterile saline by gavage for 60 days. The curcumin dose was based on a previous study [16]. After 60 days of treatment, rats from all groups were euthanized with an intraperitoneal injection of a lethal dose of anesthesia (5 times the amount used to anesthe-tize the animals) (S1 Fig).

The knee capsule was sectioned and the meniscus was isolated. The distal epiphysis of the femur containing the medial condyle cartilage was isolated and fixed with 4% paraformaldehyde in 0.1 M phosphate-buffered saline (PBS) at 7.4. The fixed specimens were decalcified in 20% formic acid dissolved in distilled water for approximately 2 days. The specimens were then routinely embedded in paraffin (Paraplast- Sigma-Aldrich cas145686-99-3) and 3 μm sections perpendicular to the articular surface of the femoral cartilage were prepared.

## Quantitative analysis

Fifty sections of each sample were used to minimize variability. Every fifteenth section from each animal was stained with hematoxylin and eosin (HE). Two trained examiners determined the cartilage thickness and chondrocyte count in a blinded manner. Histological sections were analyzed using a digital image analysis system for quantitative histomorphometry (KS-300; Carl Zeiss, Inc., Oberkochen, Germany). Each microscopic image was projected to a monitor and the cartilage area in the section was divided into five equidistant points from front to the back. Afterwards, thickness of the entire cartilage was measured perpendicularly from the surface layer to the deep layer at each determined point [17]. In each area, surface, middle or deep layers and the number of chondrocytes were determined.

## Immunohistochemistry

Histological sections (3 μm thick) were deparaffinized and rehydrated. Afterwards, sections were heated in a citrate buffer solution for antigen retrieval and incubated with methanol and a 6% hydrogen peroxide solution to quench the endogenous peroxidase activity. The sections were incubated with 1% BSA (bovine serum albumin; Sigma-Aldrich) for 45 min to block non-specific reactions. The sections were incubated with the following primary antibodies at the indicated dilutions overnight at 4°C: 1:50 SOX-5 (H-90 Santa Cruz Biotechnology), 1:200 MMP-13 (SAB4501900 Sigma-Aldrich), 1:200 MMP-8 (SAB4501895 Sigma-Aldrich), 1:100 IHH (AV45230 Sigma-Aldrich), and 1:250 Col2 (M2139 Santa Cruz Biotechnology). The Advanced HRP system (Dako, Carpinteria, Calif., USA) was used to detect antibodies and specimens were counterstained with Mayer's hematoxylin, dehydrated and mounted for observation and quantification [18].

Quantification was performed using WCIF ImageJ software. Three sections from each sample were selected, and the surface, middle, and deep sections were obtained, calibrated and color deconvolution was performed. In Vectors, H DAB was chosen and the brown image was selected. After many tests, the *Threshold* interval was selected (0 to 180) and the colored *Area Fraction* was obtained and compared to the total area of each studied section.

## Statistical analysis

Statistical analyses were performed using MATLAB® (version 9.0 R2016a, Mathworks, Inc., MA, USA) software to compare the morphometric data and the percent area displaying immunohistochemical staining in the surface, middle, and deep layers of the CG, COAG, and OAG groups. The Kolmogorov-Smirnov test was applied to evaluate the normality index of the samples from different origin, showing rejection in all cases. Therefore, the Kruskal-Wallis non-parametric test was used to compare the variables of interest. Thereafter, a *post hoc* Tukey-Kramer test was used to identify the significance of differences between groups. The significance level of $\alpha = 0.05$ was considered in all tests.

We performed the multifactorial PCA (principal component analysis) to explain the global differences in the studied variables between the three cartilage layers, surface (Surf), middle (Mid), and deep (Deep), in the OAG and COAG, and the results are shown in the

supplementary data. For this analysis, two matrices of samples were created. In the first matrix, the OAG and GOAC were set as variables and the cartilage layers, namely, Surf, Mid and Deep, were used as the PCA factors, yielding a 15x2 matrix (3 rows for each factor). The second matrix used the opposite scheme, as the cartilage layers were considered variables and the conditions OAG and COAG as PCA factors, yielding a 10x3 matrix (5 rows for each factor). In both procedures, a z-score was calculated for each column. PCA evaluates covariance clusters among all combinations, considering all variables and all trials. The clusters express the maximum similarities between each group analyzed, and groups might clearly be linearly separated.

## Results

Hematoxylin and eosin staining of the sagittal sections of articular cartilage revealed the distribution of chondrocytes, and the layers of articular cartilage (surface, middle, and deep) were differentiated. Chondrocytes in the surface layer showed an oval morphology and were organized parallel to the articular surface. Chondrocytes in the middle layer presented a more rounded morphology than that of the cells in the surface layer, and the deep layer represented the tide mark, which is defined as the transition of the noncalcified layer to the calcified layer (Fig 2A, 2B and 2C).

### Quantification of immunohistochemical staining

Col2 immunostaining was observed in the ECM of all groups and layers (Fig 2D, 2E and 2F), but the femoral articular cartilage exhibited higher Col2 expression, with significant differences observed in all cartilage layers (surface, middle, and deep), compared with that in the OAG. The CG did not show significant differences in Col2 expression in all layers compared with that in the COAG (Fig 3). The results were also confirmed using multifactorial analysis (S2 Fig).

IHH immunostaining was observed in the cytoplasm and nucleus of chondrocytes (Fig 2G, 2H and 2I). IHH expression was not significantly different between the groups in the surface and deep cartilage layers, but a significant difference in its expression in the middle layer was observed between the COAG and OAG (Fig 4). A multifactorial analysis confirmed these results (S3 Fig).

MMP-8 immunostaining was observed in the nucleus of chondrocytes (Fig 5A, 5B and 5C) and MMP-8 was expressed at a significantly higher level in the surface layer of the COAG compared with that in the OAG and in the middle layer of the COAG compared with the CG. In the deep layer, MMP-8 was expressed at significantly lower levels in the OAG than in the COAG and CG (Fig 6). These results were confirmed using a multifactorial analysis (S4 Fig).

MMP-13 immunostaining was observed in the cytoplasm and nucleus of chondrocytes in the layers of articular cartilage (Fig 5D, 5E and 5F). MMP-13 was expressed at significantly different levels in the surface layer between the OAG and COAG and in the middle and deep layers between the CG and OAG (Fig 7). These results were confirmed using a multifactorial analysis (S5 Fig).

SOX-5 immunostaining was observed in the cytoplasm and nucleus of chondrocytes (Fig 5G, 5H and 5I). Significantly higher SOX-5 expression was observed in the surface layer of the COAG compared with that in the OAG and in the deep layer of the COAG compared with the CG (Fig 8). The results were confirmed using a multifactorial analysis (S6 Fig).

### Number of chondrocytes and cartilage thickness

A significant difference in the thickness of the femoral articular cartilage was observed in the middle and deep layers. The OAG exhibited the thickest cartilage and the CG displayed the

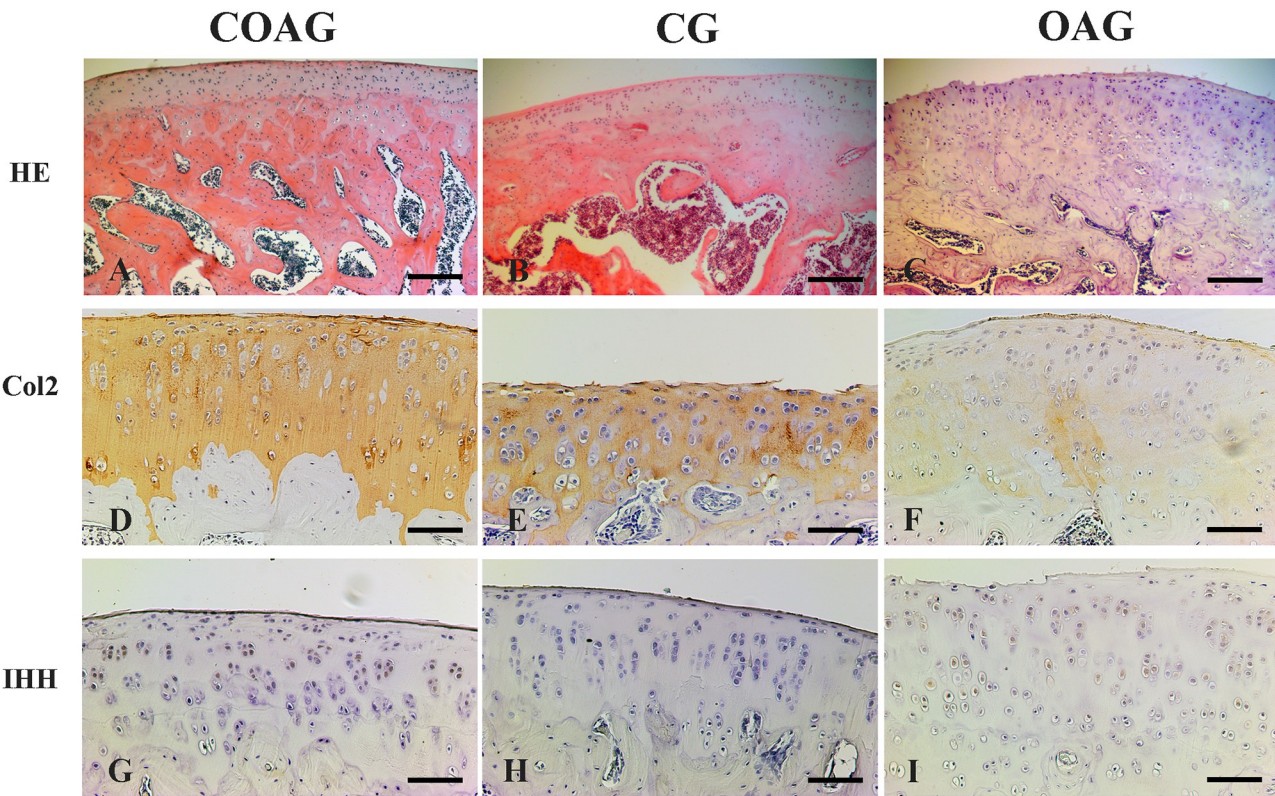

**Fig 2. Photomicrographs of histological sagittal sections of the rat femoral articular cartilage.** Images of articular cartilage from the COAG (A), CG (B), and OAG (C) stained with HE; the surface, middle, and deep layers were observed. Immunohistochemical staining for Col2 was observed in the layers (surface, middle, and deep) of articular cartilage in the COAG (D), CG (E), and OAG (F) and IHH expression was detected in the layers of articular cartilage in the COAG (G), CG (H), and OAG (I). HE scale bar: 200 μm, immunohistochemistry scale bar: 50 μm.

thinnest cartilage, with significant differences (Fig 9). These results were confirmed by performing a multifactorial analysis (S7 Fig).

The number of chondrocytes in the surface layer of the femoral articular cartilage was significantly different between the COAG and OAG. The middle layer of the CG contained fewer chondrocytes than the COAG and OAG. A significant difference in the number of chondrocytes in the deep layer was not observed between the groups (Fig 10). These results were confirmed by performing a multifactorial analysis (S8 Fig).

## Discussion

Curcumin is one of the most frequently studied natural products to date, due to its broad-spectrum preclinical activity and low oral toxicity. Curcumin is reported to be a beneficial treatment for inflammation in patients [13] by exerting protective effects on the nervous system [19], circulatory system, digestive system, and tumor metastasis [20]. In patients with OA, curcumin improves quality of life and enables a decrease in the consumption of steroidal inflammatory drugs [21]. In cells, animal models, and human studies, curcumin reduces OA inflammation [22], and one of the mechanisms identified in rabbit chondrocytes targeted advanced glycation end products [23]. Curcumin inhibits IL-1β-induced cell death, apoptosis, and the production of IL-6, IL-8, TNF-α, PGE₂, ICAM-1, and COX-2 in cultured human chondrocytes [24]. Notably, our OA model established using zymosan is based on macrophage activation and the

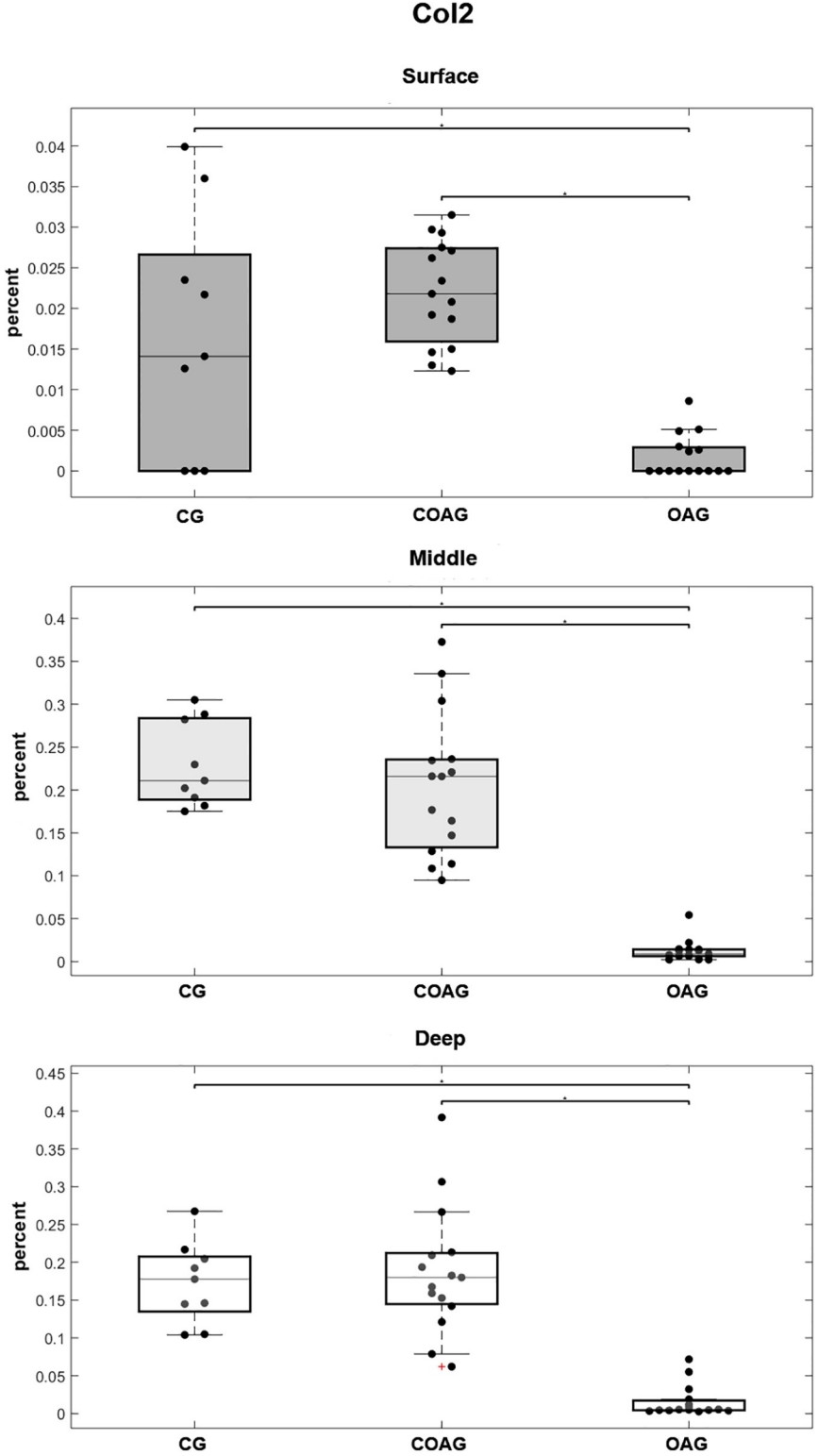

**Fig 3. Quantification of Col2 expression.** Col2 expression in the surface layer was significantly different when the CG and COAG were compared with the OAG (p = 0.0332 and p = 0.00001, respectively). Col2 was expressed at significantly different levels in the middle layer of the CG and COAG compared with that in the OAG (p = 0.00002 and p = 0.00001, respectively). Significant differences in Col2 expression in the deep layer were observed between the CG and OAG and between the COAG and OAG (p = 0.0002 and p = 0.00001, respectively); p<0.05.

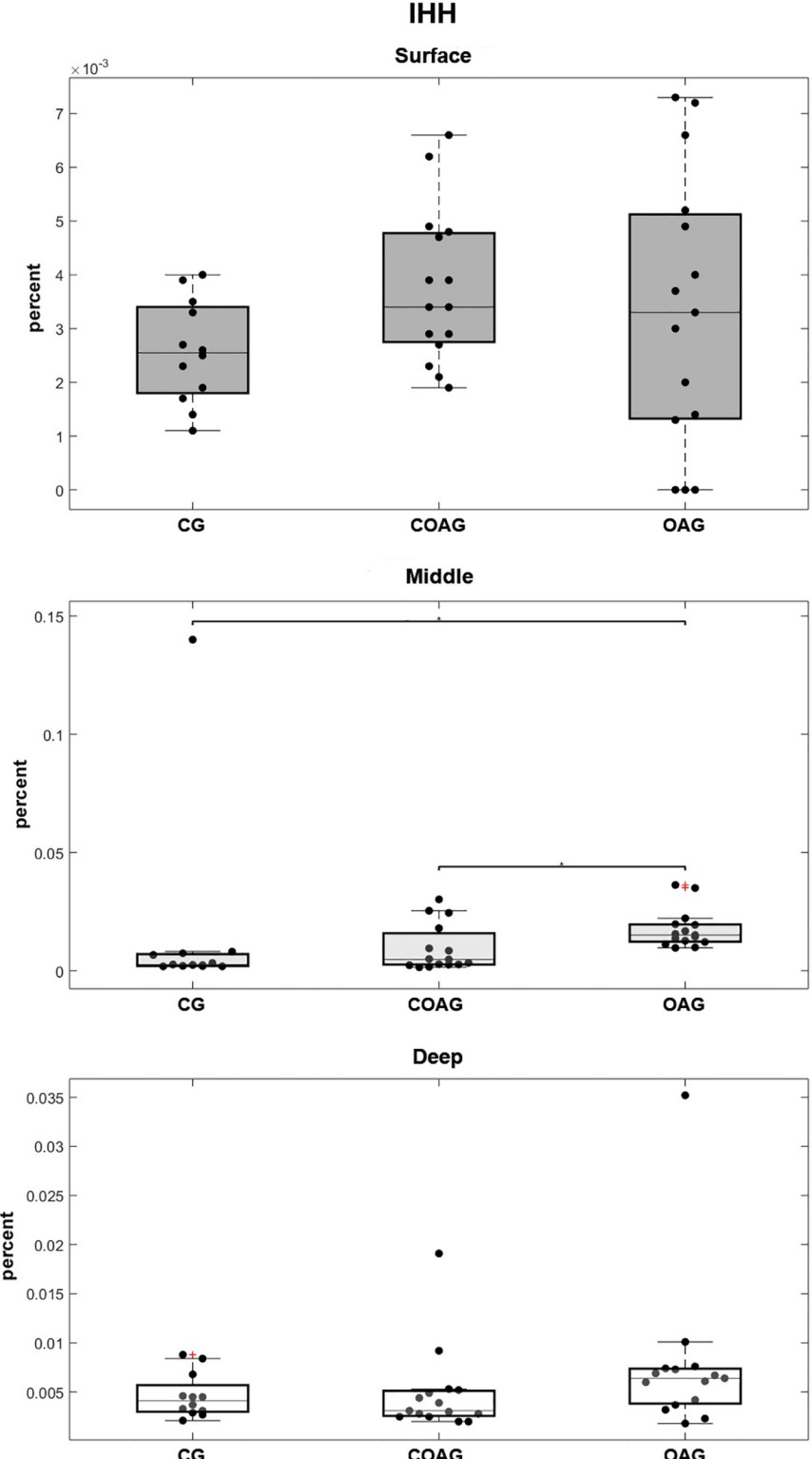

**Fig 4. Quantification of IHH expression.** Significant differences in IHH expression in the surface and deep layers were not observed between the CG, COAG and OAG. The CG and COAG displayed significant differences in IHH expression in the middle layer compared with that in the OAG (p = 0.00001 and p = 0.0202, respectively); p<0.05.

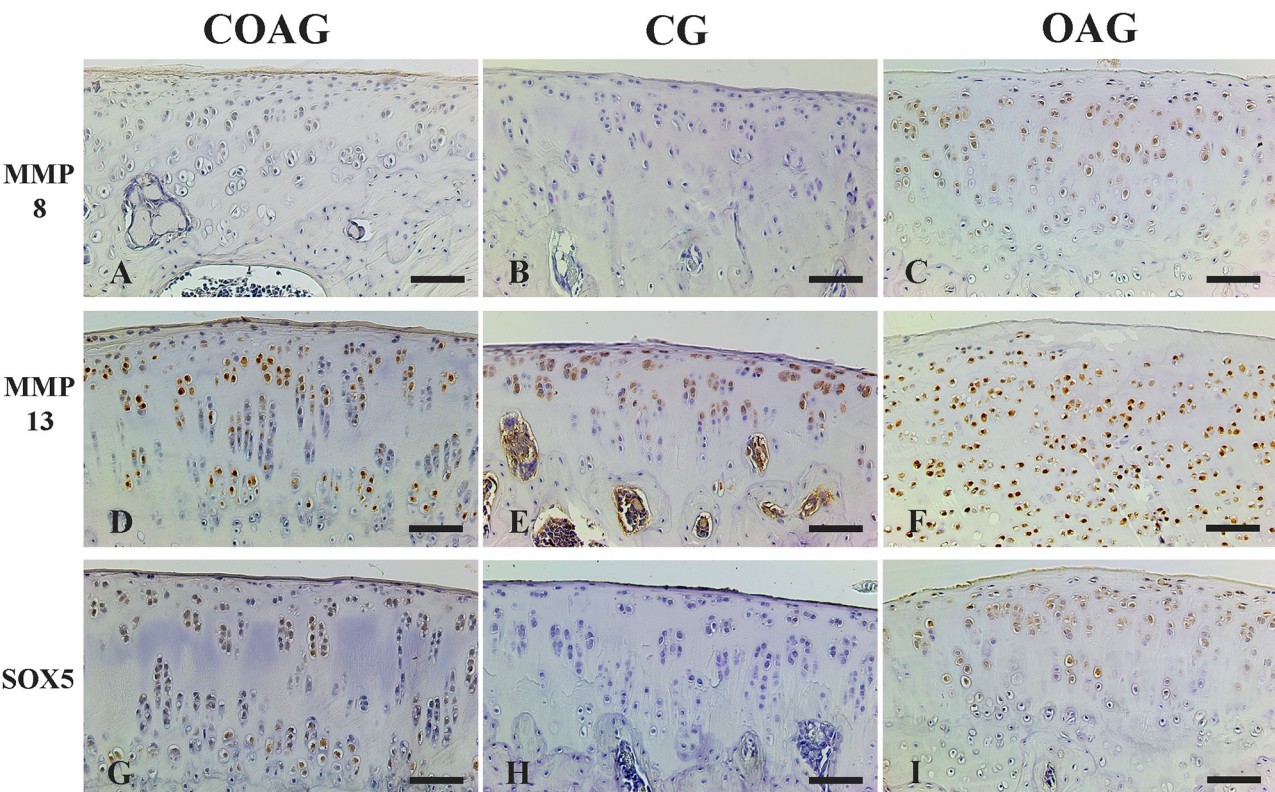

**Fig 5. Photomicrographs of immunohistochemical staining for MMP-8, MMP-13 and SOX-5.** MMP-8 expression was observed in the layers (surface, middle, and deep) of articular cartilage in the COAG (A), CG (B), and OAG (C). MMP-13 expression was detected in the layers of articular cartilage in the COAG (D), CG (E), and OAG (F). SOX-5 was expressed in the layers of articular cartilage in the COAG (G), CG (H), and OAG (I). Scale bar: 50 μm.

subsequent production of proinflammatory cytokines, arachidonate mobilization, protein phosphorylation and activation of complement via the alternative pathway [25].

According to Park et al. [26], curcumin reduces MMP-3 and MMP-13 expression in the articular cartilage of estrogen-deficient rats, preventing collagen degradation in the knee. This result contradicts the data reported in the present study, as our results revealed an increase in MMP-13 expression in the surface layer of the COAG compared with that in the OAG. As shown in other studies [23], curcumin reduces MMP-13 levels in rabbit chondrocytes and prevents the activation of nuclear factor kappaB (NF-κB). Kumar et al. [27] reported an inhibitory effect of curcumin on MMP and different cellular signaling pathways, such as NF-κB/mitogen-activated protein kinase (MAPK)/phosphoinositide 3-kinase (PI3K) and Janus kinase (JAK)/signal transduction and activation transcription (STAT). Moreover, a curcumin treatment was reported to increase Col2 synthesis, inhibit MMP-13 and reduce the degradation of extracellular matrix in chondrocytes undergoing apoptosis induced by sodium nitroprusside [28].

The present study revealed an increase in Col2 degradation in the surface, middle and deep cartilage layers in the OAG compared with those in the CG and COAG. These results corroborate molecular data for Col2 synthesis and degradation in human OA clinical research [29], in histological experiments that showed an increase in Col2 damage in OA cartilage [4,30,31] and in chondrocyte cultures showing the inhibition of Col2 synthesis [24]. An increase in Col2 expression and a decrease in MMP-13 expression was observed in rat chondrocyte cultures

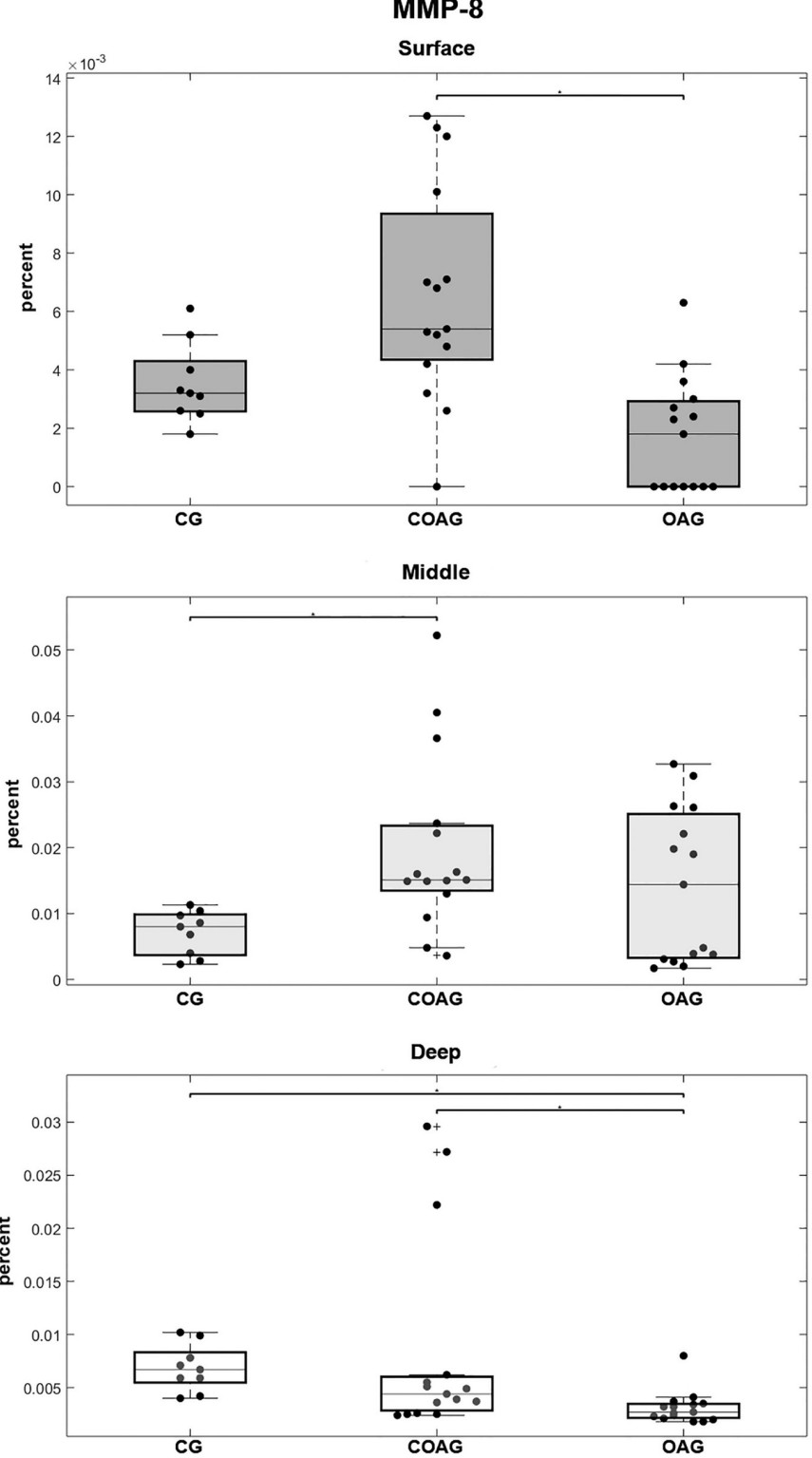

**Fig 6. Quantification of MMP-8 expression.** The COAG showed a significant difference in MMP-8 expression in the surface layer compared with that in the OAG (p = 0.0001). The CG displayed a significant difference in MMP-8 expression in the middle layer compared with that in the COAG (p = 0.0243). The CG and COAG showed significant differences in MMP-8 expression in the deep layer compared with that in the OAG (p = 0.0005 and p = 0.0293, respectively); p<0.05.

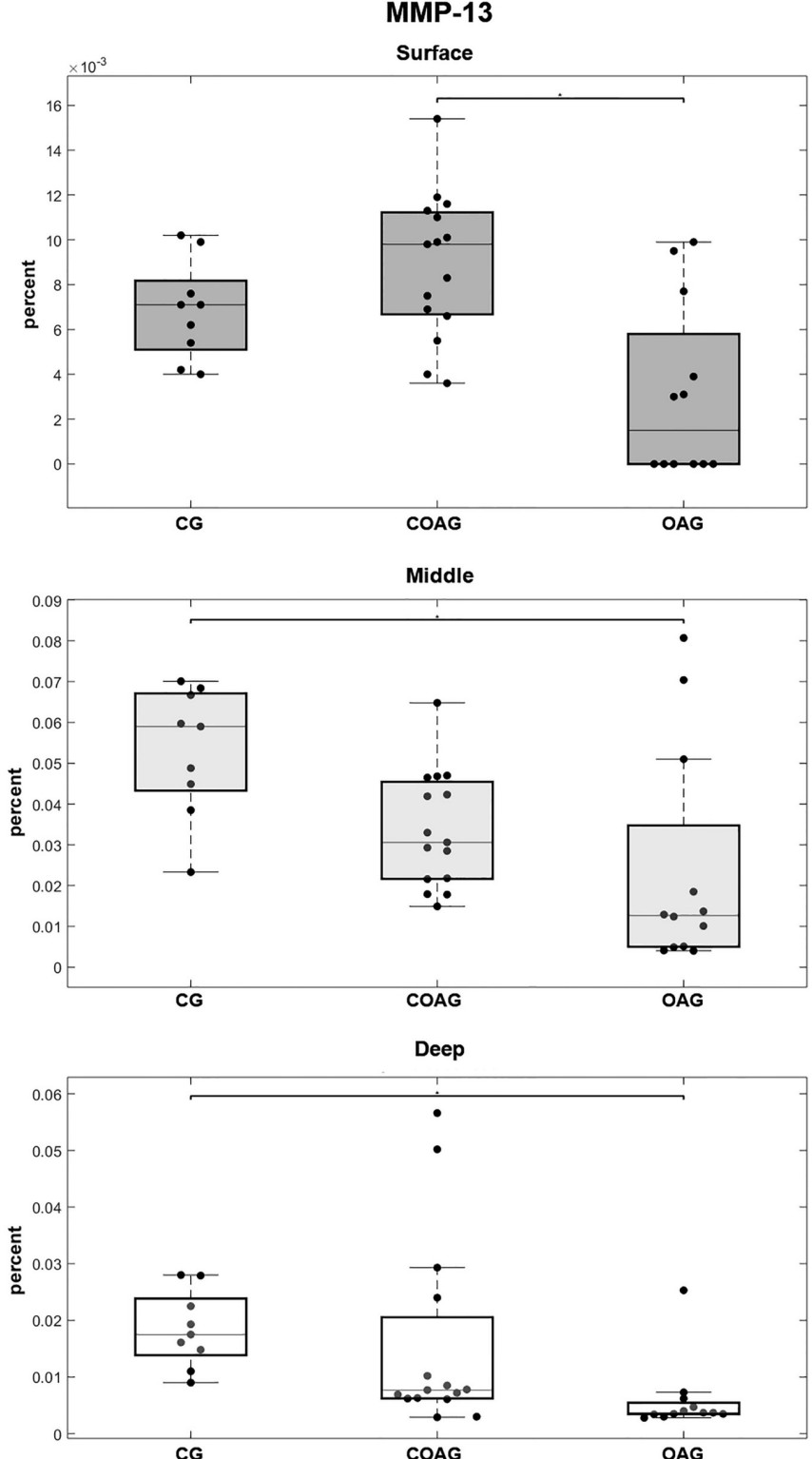

**Fig 7. Quantification of MMP-13 expression.** The COAG exhibited a significant difference in MMP-13 expression on the surface layer compared with that in the OAG (p = 0.0011). The CG showed a significant difference in MMP-13 expression in the middle and deep layers compared with that in the OAG (p = 0.0043 and p = 0.0007, respectively); p<0.05.

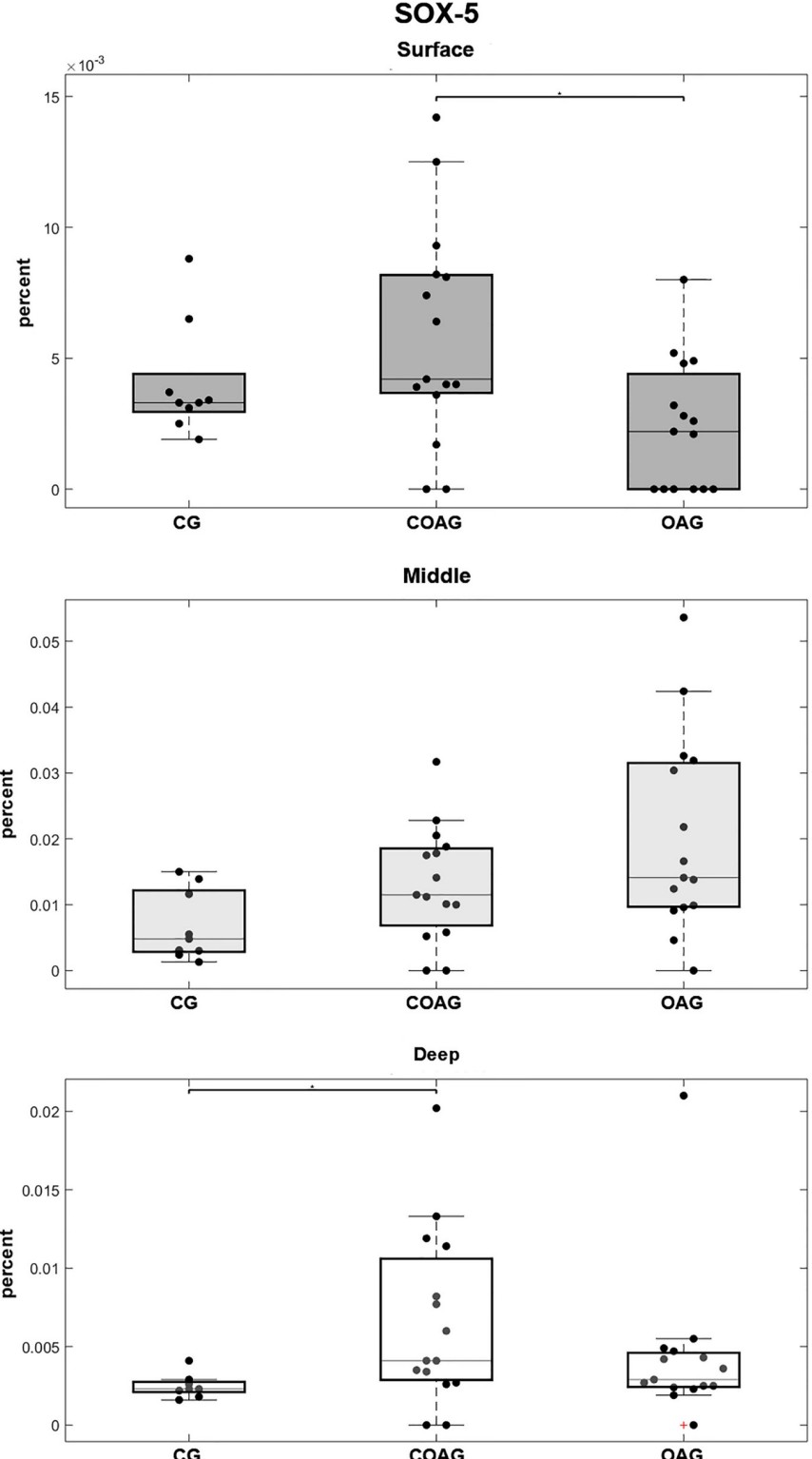

**Fig 8. Quantification of SOX-5 expression.** The COAG showed a significant difference in SOX-5 expression in the surface layer compared with that in the OAG (p = 0.0180). Significant differences in SOX-5 expression in the middle layer were not observed among the CG, COAG, and OAG. The CG displayed a significant difference in SOX-5 expression in the deep layer compared with that in the COAG (p = 0.0171); p<0.05.

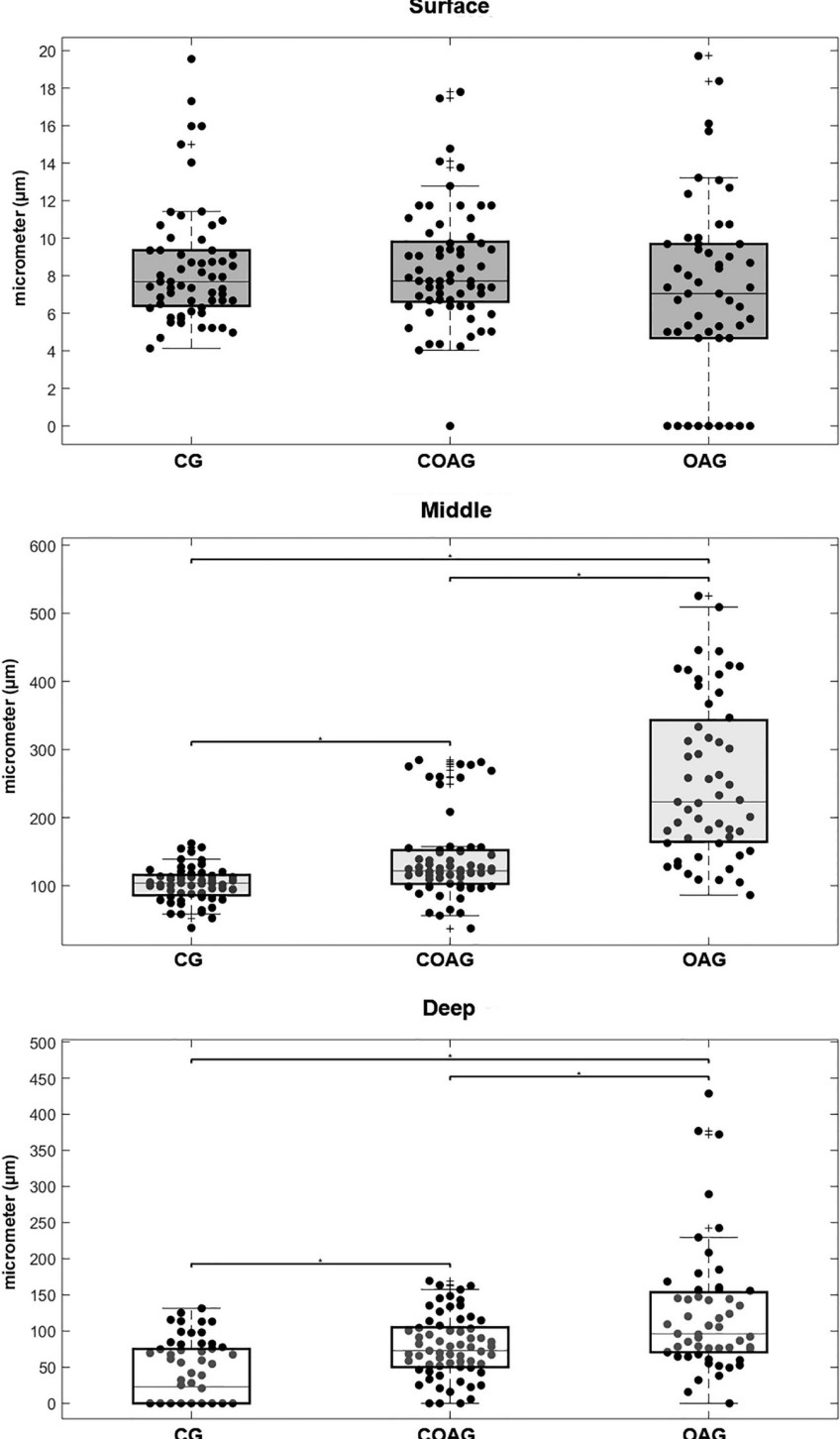

**Fig 9. Articular cartilage thickness.** Significant differences in the thickness of the articular cartilage in the surface layer were not observed among the CG, COAG, and OAG. The CG displayed a significant difference in the thickness of the middle layer compared with those in the COAG and OAG (p = 0.0009 and p = 0.00001, respectively), and the COAG showed a significant difference in the thickness of the middle layer compared with that in the OAG (p = 0.00001). The CG displayed a significant difference in the thickness of the deep layer compared with those in the

COAG and OAG (p = 0.0001 and p = 0.00001, respectively), and the COAG showed a significant difference in the thickness of the deep layer compared with that in the OAG (p = 0.0190); p<0.05.

treated with curcumin [32]. These results are similar to our data showing an increase in Col2 expression in all layers of articular cartilage in the COAG compared to that in the OAG. Hor-cajada et al. [33] reported an increase in the levels of the Coll2-1 biomarker, which is a marker of Col2 degradation, in the serum of patients with OA of the knee and hip. Moreover, the authors observed an early decrease in the levels of Coll2-1 in the group treated with rutin and curcumin. Those results corroborate our findings showing a decrease in Col2 degradation in all layers of articular cartilage in the CG and COAG compared to that in the OAG. The Col2 degradation in surface, middle, and deep layers of OA cartilage is an important finding that has also observed in some normal cartilage samples [30] and was first reported in the surface layer of the OA cartilage [4].

As shown in the study [34], when large amounts of Col2 are cleaved during normal development, chondrocyte hypertrophy and matrix mineralization are observed, along with the up-regulation of MMP-13. Moreover, human chondrocytes isolated from healthy adults constitu-tively express and secrete MMP-13, but the protein is rapidly endocytosed and degraded by chondrocytes. Based on these findings, MMP-13 may play a role in the physiological turnover of cartilage extracellular matrix [35]. Consistent with these findings, our results revealed a decrease in MMP-13 expression in the middle and deep layers of the OAG compared with that in the CG, but MMP-13 expression was decreased in the surface layer of the OAG com-pared to that in the COAG. Poole et al. [5] postulated that MMP-13 activity, collagen proteoly-sis, and chondrocyte hypertrophy are related and that the degradation of Col2 tended to induce collagenase activity mediated by MMP-1 and MMP-13, as well as IL-1 and TNFα production. Meanwhile, MMP-3 and MMP-9 are activated by IL-1β and TNF-α [36].

According to Shlopov et al. [37], MMP-13 presents low level of regulation in chondrocytes located next to OA lesions and a high level of regulation in a more distant areas, indicating that the presence of a different cytokine in the environment might modulate the effect of TGF-β. In our model, the lesion was uniformly distributed and MMP-13 was expressed at lower lev-els in the OA group than in the other groups. Cartilage degradation occurs in the presence of an imbalance between MMPs and tissue inhibitors of metalloproteinases (TIMPs) synthesized by chondrocytes [38,39,40].

In studies by Fernandes et al. [41], MMP-1 and MMP-8 were more frequently detected in the surface layer, whereas MMP-13 was observed in the deep layer of OA cartilage in dogs. Burrage et al. [2] postulated that this distribution was because MMP-1 is a product of synovial cells and MMP-8 is a product of neutrophils. These studies reinforce our findings, as the levels of MMP-8 and MMP-13 were increased in the surface and deep layers, respectively, of the OAG but were present in all layers of the OAG. Tetlow et al. (2001) [42] reported high levels of MMP-1, MMP-8 and MMP-13 in the surface layer of cartilage from patients with OA. In the same study, MMPs 1, 8, and 13 were associated with areas that exhibited matrix depletion, fibrillation, chondrocyte clusters, and a loss of metachromasia. In the present study, different results were obtained, as MMP-8 was expressed at higher levels in the surface layer of the CG and COAG than in the OAG. On the other hand, our results are consistent with the findings reported by Arend et al. (1995) [43] and Lotz et al. (1995) [44], who showed that chondrocytes under normal or pathological conditions exhibit differences in the synthesis of the proinflam-matory cytokines IL-1 and TNF, as well as MMPs.

The present study described MMP-8 expression in the CG. According to Lin et al. [45], MMP-8 cleaves the main collagens present in cartilage, such as collagens 1, 2 and 3, other

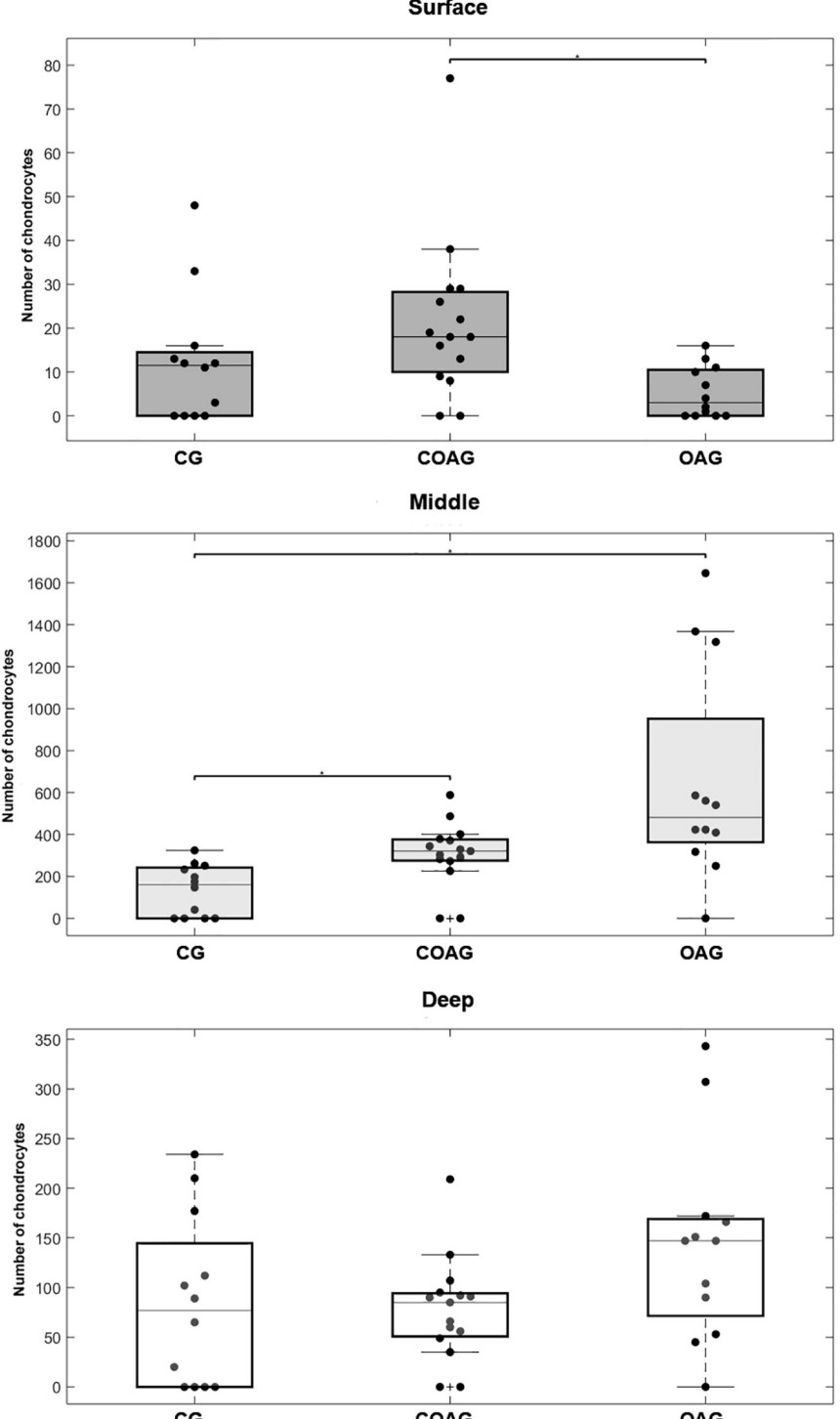

**Fig 10. Chondrocyte counts in each layer.** The COAG showed a significantly different number of chondrocytes in the surface layer compared with that in the OAG (p = 0.0097). A significant difference in the number of chondrocytes in the middle layer was observed between the CG and the COAG and OAG (p = 0.0366 and p = 0.0002, respectively). Significant differences in the numbers of chondrocytes in the deep layer were not observed between the CG, COAG, and OAG; p<0.05.

components of extracellular matrix, non-structural molecules, and affect neutrophil migration. On the other hand, Billinghurst et al. [31] reported an association between increased MMP-8 expression and an increase in Col2 cleavage in samples of human cartilage with osteoarthritis. Our results differed from previous studies because MMP-8 and Col2 were expressed at higher levels in the surface layer of the COAG than in the OAG; MMP-8 and Col2 were also expressed at higher levels in the deep layer of the CG and COAG than in the OAG. Some researchers detected MMP-8 expression in neutrophils, chondrocytes, mature chondrocytes, and chondrocytes exposed to pro-inflammatory stimuli [46].

The findings from the present study identified important changes in SOX-5 expression, as a significant increase in the expression of this transcription factor was observed in the surface and deep layers of the COAG compared with those in the OAG and CG. Lee and Im [47] analyzed three SOX proteins (5, 6 and 9) and concluded that they not only promoted the healing of osteochondral defects but also prevented the progression of surgically induced osteoarthritis in rats, as evidenced by a significant increase in the levels of glycosaminoglycans and expression of the Col2 mRNA and protein.

Khan et al. [48] analyzed the levels of SOX-5, 6 and 9, among other markers, in stem cells derived from infrapatellar fat cushion in articular cartilage defects of the knee in humans. An increase in the expression of these three SOX proteins was observed in response to hypoxia, prompting the authors to conclude that chondrogenesis increased in culture under these conditions. Findings from the present study did not corroborate the results obtained by Kim and Im [49], who evaluated the expression of these three SOX mRNAs and proteins in femoral cartilage from rats affected by OA. A decrease in the expression of the SOX mRNAs and an increase in SOX protein levels were observed. These researchers suggested that the high levels of these proteins represented a delayed effect on gene expression. In our study, SOX-5 expression was increased in the surface and deep layers of the COAG compared with that in the OAG and CG. On the other hand, Lee and Im [50] reported a decrease in the levels of these three SOX proteins in cartilage from patients with OA.

In a study performed by Velusami et al. [24], treatment with a polar extract of curcumin did not increase chondrocyte proliferation in culture, but it reduced apoptosis and cell death. As shown in the study by Feng et al. [51], curcumin significantly attenuated the degradation of knee articular cartilage and decreased chondrocyte apoptosis in rat models that underwent anterior cruciate ligament surgery in a dose-dependent manner. Our results may be complementary, as we showed an increased number of chondrocytes in the surface layer of the COAG compared to that in the OAG as well as an increased number of chondrocytes in the middle layer of the OAG and COAG compared to that in the CG. According to Cao et al. [52], curcumin inhibits chondrocyte hypertrophy by decreasing IHH expression. Moreover, curcumin also reduces matrix degradation in the cartilage by balancing the anabolic and catabolic activity of repair cells [53],inhibits the effects of sodium nitroprusside (SNP) on inducing chondrocyte apoptosis, and maintains the balance of extracellular matrix metabolism [28].

The present study revealed a significant difference in IHH expression in the middle layer of the OAG compared with those in the CG and COAG. Similar results were reported by Zhong et al. [54], who related the progression of OA to the expression of IHH. Van der Kraan and Van den Berg [55] suggested a correlation between IHH expression and the progression of OA in animal models, due to protease-mediated effects on the progression of OA. As shown in the study by Cao et al. [52], curcumin effectively reduces IHH expression in chondrocytes and mesenchymal stem cells, inhibiting GLI2 gene expression and increasing GLI3 expression to subsequently modulate cartilage homeostasis and maintain chondrocyte phenotypes, which corroborated the histological findings from the present study.

According to Bhosale and Richardson [56], four cartilage layers are detected in the knee, namely, the surface, middle, deep, and calcified layers. In the same study, chondrocytes synthesized high concentrations of collagen and low concentrations of proteoglycans in the surface layer, while higher concentrations were observed in the middle and deep layers. Chondrocyte hypertrophy and increased chondrocyte proliferation have been observed in the junction between the synovium and cartilage in mice with zymosan-induced arthritis [57]. Meanwhile Mankin [58] reported micro injuries in chondrocytes, cellular degeneration and cellular death induced by impacts on the surface of joints or repetitive loading in articular cartilage. Moreover, these types of damage potentially disrupt the collagen matrix, leading to increases in hydration, cartilage fissures, and subchondral bone thickness. These data corroborated the findings from our study, as the cartilage thickness was increased in the middle and deep layers of the OAG compared with those in the CG and COAG.

## Conclusions

The curcumin treatment did not increase the cartilage thickness, but the treatment increased the number of chondrocytes compared with that in the osteoarthritis group. The curcumin treatment increased Col2, SOX-5, MMP-8 and MMP-13 expression but decreased IHH expression. According to our data, curcumin appears to exert a cartilage protective effect. However, additional studies are needed to obtain a better understanding of the balance between degeneration markers and clinically heathy cartilage.

## Supporting information

**S1 Fig. Flow chart of the study design.** It demonstrates periods of rest, treatments and euthanasia.
(TIF)

**S2 Fig. Cluster analysis of COL2 expression using principal component analysis (PCA).**
**(A)** The left upper quadrant shows the data distribution when considering the COAG and OAG as like variables and three cartilage layers as variables: surface (Surf), middle (Mid), and deep (Deep). The samples associated with the Surf cartilage layer displayed less variance and a high level of autosimilarity, while the Mid and Deep cartilage layers exhibited high variance and were indistinguishable. Moreover, the Surf sample cluster was linearly separated from the other clusters, emphasizing that the Surf factor had different statistical properties than the other factors. **(B)** The left lower quadrant shows the increased variance of the COAG variable in the PCA cluster. **(C)** The right upper quadrant shows the data distribution into the cartilage layers, Surf, Mid, and Deep, as variables of PCA and COAG and OAG as variable factors to evaluate the symmetry of the data. The samples from the OAG displayed less variance and a high level of autosimilarity, while the samples from the COAG exhibited a high level of variance. OAG and COAG clusters were also linearly separated, enforcing the statistically significant differences between these two groups. **(D)** The right lower quadrant show the high level of variance in the Mid and Deep variables in the PCA cluster, corroborating the findings from the inverse analysis.
(TIF)

**S3 Fig. Cluster analysis of IHH expression using principal component analysis (PCA). (A)** The left upper quadrant shows the data distribution when considering the COAG and OAG as like variables and the three cartilage layers as variables: surface (Surf), middle (Mid), and deep (Deep). The samples associated with the Mid cartilage layer displayed a high level of variance and less autosimilarity, while Surf and Deep cartilage layers exhibited less variance and were

indistinguishable. Moreover, the Mid sample cluster was linearly separated from the other factors, indicating a difference in statistical properties compared with the other factors. **(B)** The left lower quadrant shows the increased variance in the OAG variable in the PCA cluster. **(C)** The right upper quadrant shows the data distribution when considering the cartilage layers Surf, Mid, and Deep as variables of PCA and COAG and OAG as variable factors to evaluate the symmetry of the data. The samples from the COAG displayed less variance and a high level of autosimilarity, while the OAG samples exhibited a high level of variance. COAG and OAG clusters were not linearly separated, confirming the same statistical relationship between these two groups. **(D)** The right lower quadrant shows the high level of variance in the Mid variable in the PCA cluster, corroborating the results from the inverse analysis.
(TIF)

**S4 Fig. Cluster analysis of MMP-8 expression using principal component analysis (PCA).**
**(A)** The left upper quadrant shows the data distribution when considering the COAG and OAG as like variables and the three cartilage layers as variables: surface (Surf), middle (Mid), and deep (Deep). The samples associated with the Mid cartilage layer exhibited a high level of variance and less autosimilarity, while the Surf and Deep cartilage layers displayed less variance and were indistinguishable. Furthermore, the Mid sample cluster were not linearly separated from the other clusters, emphasizing that the Mid factor had same statistical properties as the other factors. **(B)** The left lower quadrant shows the increased variance of the COAG variable in the PCA cluster. **(C)** The right upper quadrant shows the data distribution when considering the cartilage layers Surf, Mid, and Deep as variables in PCA and COAG and OAG as variable factors to evaluate the symmetry of the data. The samples from the OAG displayed less variance and were indistinguishable from the COAG. The OAG and COAG clusters were also not linearly separated, confirming the same statistical relationship between these two groups. **(D)** The right lower quadrant shows the high level of variance of the Mid variable, followed by Deep and Surf in the PCA cluster, corroborating the results of the inverse analysis.
(TIF)

**S5 Fig. Cluster analysis of MMP-13 expression using principal component analysis (PCA).**
**(A)** The left upper quadrant shows the distribution of data when considering the groups COAG and OAG as like variables and the three cartilage layers as variables: surface (Surf), middle (Mid), and deep (Deep). The samples associated with the Mid cartilage layer displayed a high level of variance and less autosimilarity, while Surf and Deep cartilage layers displayed less variance and were indistinguishable. Moreover, the Mid sample cluster were not linearly separated from the other clusters, indicating that the Mid factor had the same statistic as the other factors. **(B)** The left lower quadrant shows the increased variance of the OAG variable in the PCA cluster. **(C)** The right upper quadrant shows the data distribution when considering the cartilage layers Surf, Mid, and Deep as variables of PCA and COAG and OAG as variable factors to evaluate the symmetry of the data. The samples from the COAG displayed less variance and more autosimilarity, while the OAG samples displayed a high level of variance. COAG and OAG clusters were also not linearly separated, suggesting the same statistical relationship between these two groups. **(D)** The right lower quadrant shows the high level of variance of the Mid variable in the PCA cluster, corroborating the results of the inverse analysis.
(TIF)

**S6 Fig. Cluster analysis of SOX-5 expression using principal component analysis (PCA).**
**(A)** The left upper quadrant shows the data distribution when considering the COAG and OAG as like variables and the three cartilage layers as variables: surface (Surf), middle (Mid), and deep (Deep). The samples associated with the Mid cartilage layer displayed a high level of

variance and less autosimilarity, while Surf and Deep cartilage layers exhibited less variance and were indistinguishable. Furthermore, the Mid samples were linearly separated from the other clusters, emphasizing that the Mid factor had different statistics than the other factors. **(B)** The right lower quadrant showed the high level of variance in the OAG variable in the PCA cluster. **(C)** The right upper quadrant shows the data distribution when considering the cartilage layers, Surf, Mid, and Deep, as variables in PCA and COAG and OAG as variable factors to evaluate the symmetry of the data. The samples from the COAG displayed less variance and a high level of autosimilarity, while the samples from the OAG had a high level of variance. The COAG and OAG clusters were not linearly separated, revealing the same statistical relationship between these two groups. **(D)** The right lower quadrant shows the high level of variance in the Mid variable in the PCA cluster, corroborating the results of the inverse analysis. (TIF)

**S7 Fig. Cluster analysis of the articular cartilage thickness using principal component analysis (PCA). (A)** The left upper quadrant shows the data distribution when considering the COAG and OAG as like variables and the three cartilage layers as variables: surface (Surf), middle (Mid), and deep (Deep). The samples associated with the Surf cartilage layer displayed less variance and a high level of autosimilarity, while the Deep cartilage layer displayed a high level of variance and was indistinguishable from the Mid layer. Furthermore, the Surf sample cluster were linearly separated from the other clusters, emphasizing that the Surf factor had different statistical properties from the other factors. **(B)** The left lower quadrant shows the increased variance of the OAG variable in the PCA cluster. **(C)** The right upper quadrant shows the data distribution when considering the cartilage layers Surf, Mid, and Deep as variables of PCA and COAG and OAG as variable factors to evaluate the symmetry of the data. The samples from the COAG displayed less variance and were indistinguishable from the OAG. COAG and OAG clusters were not linearly separated, confirming the same statistical relationship between these two factors. **(D)** The right lower quadrant shows the high level of variance of the Mid variable, followed by the Deep and Surf variables in the PCA cluster, corroborating the results of the inverse analysis. (TIF)

**S8 Fig. Cluster analysis of the number of chondrocytes using principal component analysis (PCA). (A)** The left upper quadrant shows the data distribution when considering the COAG and OAG as like variables and the three cartilage layers as variables: surface (Surf), middle (Mid), and deep (Deep). The samples associated with the Surf cartilage layer displayed less variance and a high level of autosimilarity, while the Deep cartilage layer showed less variance and a high level of autosimilarity compared with those in the Middle cartilage layer. Moreover, the Surf, Mid, and Deep sample clusters were linearly separated from the other clusters, emphasizing that the Surf, Mid, and Deep factors possessed different statistical properties than the other factors. **(B)** The left lower quadrant shows the increased variance of the OAG variable in the PCA cluster. **(C)** The right upper quadrant shows the data distribution when considering the cartilage layers Surf, Mid, and Deep as variables of PCA and COAG and OAG as variable factors to evaluate the symmetry of the data. The samples from the COAG displayed less variance and were indistinguishable from the OAG. COAG and OAG clusters were not linearly separated, confirming the same statistical relationship between these two factors. **(D)** The right lower quadrant shows the high level of variance of the Mid variable, followed by the Deep and Surf variables in the PCA cluster, corroborating the results of the inverse analysis. (TIF)

## Acknowledgments

We are thankful to Drᵃ. Adriana da Costa Neves from Instituto Butantan for providing technical assistance with the immunohistochemical procedures.

## Author Contributions

**Conceptualization:** Tiago Nicoliche, Diogo Correa Maldonado, Marcelo Cavenaghi Pereira da Silva.

**Data curation:** Jean Faber.

**Formal analysis:** Tiago Nicoliche, Diogo Correa Maldonado, Marcelo Cavenaghi Pereira da Silva.

**Funding acquisition:** Marcelo Cavenaghi Pereira da Silva.

**Investigation:** Tiago Nicoliche, Diogo Correa Maldonado.

**Methodology:** Tiago Nicoliche, Diogo Correa Maldonado, Jean Faber, Marcelo Cavenaghi Pereira da Silva.

**Supervision:** Marcelo Cavenaghi Pereira da Silva.

**Validation:** Tiago Nicoliche, Jean Faber.

**Writing – original draft:** Tiago Nicoliche, Diogo Correa Maldonado, Marcelo Cavenaghi Pereira da Silva.

**Writing – review & editing:** Tiago Nicoliche, Diogo Correa Maldonado, Marcelo Cavenaghi Pereira da Silva.

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
