## [Decision Letter · Decision Letter 0]

8 Nov 2019

PONE-D-19-17705

Evaluation of rat knee articular cartilage with induced arthritis and treated with Curcumin

PLOS ONE

Dear Mr. Nicoliche,

Thank you for submitting your manuscript to PLOS ONE. After careful consideration, we feel that it has merit but does not fully meet PLOS ONE’s publication criteria as it currently stands. Therefore, we invite you to submit a revised version of the manuscript that addresses the points raised during the review process.

We would appreciate receiving your revised manuscript by Dec 15 2019 11:59PM. To enhance the reproducibility of your results, we recommend that if applicable you deposit your laboratory protocols in protocols.io, where a protocol can be assigned its own identifier (DOI) such that it can be cited independently in the future. For instructions see: http://journals.plos.org/plosone/s/submission-guidelines#loc-laboratory-protocols

We look forward to receiving your revised manuscript.

Kind regards,

Chi Zhang

Academic Editor

PLOS ONE

Journal Requirements:

1. Please include the following information relating to animal experiments in your Methods section: The frequency of animal monitoring, including the specific criteria you used to monitor animal health.

2. Please also justify the chosen dose of curcumin administered to the animals.

https://doi.org/10.1007/s00441-013-1658-y

In your revision ensure you cite all your sources (including your own works), and quote or rephrase any duplicated text outside the Methods section. Further consideration is dependent on these concerns being addressed

4. We suggest you thoroughly copyedit your manuscript for language usage, spelling, and grammar. If you do not know anyone who can help you do this, you may wish to consider employing a professional scientific editing service.  

Reviewers' comments:

Reviewer's Responses to Questions

**Comments to the Author**

1. Is the manuscript technically sound, and do the data support the conclusions?

Reviewer #1: Yes

2. Has the statistical analysis been performed appropriately and rigorously? 

Reviewer #1: I Don't Know

3. Have the authors made all data underlying the findings in their manuscript fully available?

Reviewer #1: Yes

4. Is the manuscript presented in an intelligible fashion and written in standard English?

Reviewer #1: Yes

5. Review Comments to the Author

Reviewer #1: “Evaluation of rat knee articular cartilage with induced arthritis and treated with Curcumin” by Tiago Nicoliche et al. attempts to evaluate the anti-inflammatory effects of curcumin in the induced osteoarthritis treatment in rats’ knee. I recognize the effort made or the authors to develop this histologic descriptive study about some keys elements of OA development. Conceptually this manuscript is very similar to other manuscripts published (Mol Med Rep. 2017 Aug;16(29); Osteoarthritis Cartilage. 2015 Jan;23(1):94-102; Oxid Med Cell Longev. 2019 May 16;2019; Biosci Rep. 2018 Jul 2;38(4); Mol Med Rep. 2017 Aug;16(29). So, what is the novelty of this study? I do not think that this manuscript adds significantly to what is already know and is not novel enough to warrant a new full length publication. I have the following suggestions for improvement.

Major comments:

1) The style of the manuscript should be improved. The discussion reads like a review at certain places. It should be more focused, discussing the experimental findings in light of the available data in the literature as they relate to the development of osteoarthritis. “In studies carried out by Fernandes et al. … Burrage et al. (2006) believed that … Tetlow et al. (2001) (40) found …” This paragraph reads like a list without concluding what is unknown, for example.

2) As it is suggested that curcumin could be administered as part of a dietary supplementation aimed at preventing inflammation and reducing OA severity, it is important to fully analyse the effects of the tested dose on control animals in addition to those subjected to the OA model. Does orally administered curcumin also improved Col2, IHH, SOX-5 expressions and the number of chondrocytes in unchallenged joints or in naturally ageing animals?

3) Does the curcumin lead to any reduction in synovial tissue inflammation? At least the presentation of wider field images of the joint including synovial and cartilage would help in understanding the therapeutic impact of the treatment.

4) Please provide more specific reasons why intraarticular injection of Zymosan is a proper model of OA, since the mechanism of OA is not fully understood

5) Is there clinical report or field study showing that people regularly consuming diet containing curcumin have lower rate of OA compared to those not consuming such diet?

Minor comments:

1) The Schedule of OA induction and treatment it is not clear. Animals were euthanized 8 weeks after induction of OA? or 8 weeks after curcumin was daily administered for 60 days?

2) The results should be described in the same order than results are presented in figures.

3) A figure describing possible model for how curcumin modulate inflammation could be good.

6. PLOS authors have the option to publish the peer review history of their article (what does this mean?). If published, this will include your full peer review and any attached files.

Reviewer #1: No

---

## [Author Response · Author response to Decision Letter 0]

27 Jan 2020

It was sent a doc file in the previous steps responding all the questions.

---

## [Decision Letter · Decision Letter 1]

26 Feb 2020

Evaluation of the articular cartilage in the knees of rats with induced arthritis treated with curcumin

PONE-D-19-17705R1

Dear Dr. Nicoliche,

We are pleased to inform you that your manuscript has been judged scientifically suitable for publication and will be formally accepted for publication once it complies with all outstanding technical requirements.

With kind regards,

Chi Zhang

Academic Editor

PLOS ONE

---

## [Editor Report · Acceptance letter]

2 Mar 2020

PONE-D-19-17705R1 

Evaluation of the articular cartilage in the knees of rats with induced arthritis treated with curcumin 

Dear Dr. Nicoliche:

I am pleased to inform you that your manuscript has been deemed suitable for publication in PLOS ONE. Congratulations! Your manuscript is now with our production department. 

With kind regards,

on behalf of

Dr. Chi Zhang 

Academic Editor

PLOS ONE